# On the Below- and Aboveground Phenology in Deciduous Trees: Observing the Fine-Root Lifespan, Turnover Rate, and Phenology of *Fagus sylvatica* L., *Quercus robur* L., and *Betula pendula* Roth for Two Growing Seasons

Bertold Mariën [1],*, Ivika Ostonen [2], Alice Penanhoat [3], Chao Fang [1,4], Hòa Xuan Nguyen [1], Tomáš Ghisi [5], Páll Sigurðsson [6], Patrick Willems [7] and Matteo Campioli [1]

1   PLECO (Plants and Ecosystems), Department of Biology, University of Antwerp, 2610 Antwerp, Belgium; fangchao567@gmail.com (C.F.); hoanguyen1159@gmail.com (H.X.N.); matteo.campioli@uantwerpen.be (M.C.)
2   Department of Geography, Institute of Ecology and Earth Sciences, University of Tartu, 51003 Tartu, Estonia; ivika.ostonen@ut.ee
3   CBL (Centre of Biodiversity and sustainable Land Use), Department of Silviculture and Forest Ecology of the Temperate Zones, University of Göttingen, 37077 Göttingen, Germany; alice.penanhoat@uni-goettingen.de
4   School of Applied Meteorology, Nanjing University of Information Science & Technology, Nanjing 210004, China
5   CzechGlobe, Department of Climate Change Impacts on Agroecosystems, Czech Academy of Sciences, 61300 Brno, Czech Republic; tom.ghisi@gmail.com
6   Department of Forestry, Agricultural University of Iceland, 311 Borgarnes, Iceland; palls@lbhi.is
7   Hydraulics and Geotechnics Section, Department of Civil Engineering, KU Leuven, Kasteelpark Arenberg 40, 3001 Leuven, Belgium; patrick.willems@kuleuven.be
*   Correspondence: bertold.marien@uantwerpen.be; Tel.: +32-65-9333

**Abstract:** We tested the relation between the below- and aboveground tree phenology, determining if beech and oak have a greater fine-root lifespan and a smaller turnover rate than birch and if thinner fine-roots or fine-roots born in spring have a shorter lifespan and greater turnover rate than thicker fine-roots or fine-roots born in another season. The fine-root phenology, bud burst, and leaf senescence in Belgian stands were monitored using minirhizotrons, visual observations, and chlorophyll measurements, respectively. The fine-root phenology and the lifespan and turnover rate were estimated using generalized additive models and Kaplan–Meier analyses, respectively. Unlike the aboveground phenology, the belowground phenology did not show a clear and repeating yearly pattern. The cumulative root surface remained stable for birch but peaked for beech and oak around summer to autumn in 2019 and spring in 2020. The new root count was larger in 2019 than in 2020. The mean lifespan of fine-roots with a diameter below 0.5 mm (308 to 399 days) was shorter than those with a diameter between 0.5 to 1 mm (438 to 502 days), 1 to 2 mm (409 to 446 days), or above 2 mm (418 to 471 days). Fine-roots born in different seasons showed a species-specific lifespan and turnover rate.

**Keywords:** belowground phenology; European beech; fine-root lifespan; fine-root turnover rate; pedunculate oak; silver birch

## 1. Introduction

### 1.1. The Functions of Fine-Roots

Ideally, roots should be defined based on their functional characteristics [1]. However, for pragmatic reasons and because of unofficial convention, roots are usually defined on a morphological basis. For example, roots with a diameter below 2 mm are usually defined as fine-roots, while lignified (i.e., woody) roots with a diameter above 2 mm and secondary development are usually defined as coarse roots [2]. Fine-roots are therefore

considered to be non-woody, short-lived, and of main importance for a tree's resource uptake (often mediated through microbial interaction), while coarse roots are considered to live longer and be of main importance for a tree's stability and resource transportation [3–5]. Nevertheless, certain fine-roots (i.e., absorptive or transport roots when categorized by their functional characteristics) can also perform a resource-transportation function. In fact, recent research suggests that fine-roots with a diameter below 2 mm contain a large portion of lignified woody roots with secondary development. Roots with a diameter below 0.5 mm may then be considered mostly absorptive roots with a primary or secondary structure, while roots with a diameter between 0.5 and 2 mm may largely be considered woody transport roots [6,7]. Roots of the same diameter sometimes have a different branching structure (i.e., primary or secondary roots) or function [2,8,9]. Apart from resource uptake, resource transportation, and physical stabilization, roots can also store nutrients and carbohydrates or act as sensors for exterior conditions [10–13].

### 1.2. Contribution of Fine-Roots to Forest Biomass and Production

Fine-roots can add nutrients and carbon into the soil through their continuous growth and mortality [10,11]. This process of fine-root litter production is often measured using the fine-root turnover rate, or the inverse of the median root lifespan, and can be interpreted as the number of times in which the fine-root biomass is replaced within a given timeframe [3]. In fact, the fine-root turnover is estimated to account for approximately 10% to 60% of the annual net primary production (NPP) or 22% to 63% of the gross primary production (GPP) in forests globally [1,5,8,14–25].

Holmes and Likens [13] showed that roots typically represent approximately 20% to 25% of the total plant biomass in forests. The largest percentage of the root biomass would here be represented by woody roots, while fine-roots would represent only around 3% to 10% of the total plant biomass [13,26–28]. More recent estimates of Huang et al. [29] showed that around 12% of the total global root biomass is present in temperate broadleaf forests. In these forests, most root biomass is found in the surface soil horizon: an estimated 83% is present in the first 20 cm and 40% in the organic horizon [13,30]. Although changes are possible throughout the forest development, the root-to-shoot biomass ratio for mature temperate broadleaf forests would, with an estimated value of 0.24, also lie very close to the global terrestrial average root-to-shoot biomass ratio estimated around $0.25 \pm 0.10$ [13,29,31].

### 1.3. The Fine-Root Lifespan, Turnover Rate, and Phenology of Temperate Deciduous Trees

The fine-root lifespan and turnover rate affect the timely uptake of resources, the competition among species, and the carbon fluxes in the soil [19]. It is therefore of paramount importance that we understand the coupling between the fine-root lifespan and the turnover rate (and the fine-root phenology). For example, due to the difference in carbohydrate reserve content, the lifespan and the turnover rate of the fine-roots may differ per season of fine-root birth. Fine-roots born in spring may thus live shorter lives than those born later [8,32]. The literature reports a wide variation in the fine-root phenological patterns among species and years. Observed patterns in the fine-root phenology throughout one year are linked to the timely availability of resources and may thus include: a constant fine-root production, a variable fine-root production with one peak, or a variable fine-root production with multiple peaks (usually two) [19,33–35]. Deciduous trees in temperate regions usually have a variable fine-root production with values that peak in June in the Northern Hemisphere, remaining relatively constant throughout summer and decreasing later on [1,36]. However, drought and heat stress may cease the fine-root production [1,37,38].

Relatively little information is available on the phenology of fine-roots in deciduous trees [39]. Therefore, it is no surprise that even less attention is dedicated to the coupling between the below- and aboveground phenological processes in deciduous trees. In fact, most information on this coupling is related to the differences in resource acquisition determined

by leaf or root traits, rather than by differences in the actual phenology [40,41]. Whenever inferences are made between the below- and aboveground phenology in deciduous trees, these are almost exclusively drawn between phenological patterns in the fine-roots and the bud burst, while usually neglecting the leaf senescence period. In addition, these inferences between above- and belowground phenology are usually drawn using data of relatively low resolution (i.e., approximately once a month) [15,19,42]. Recent technologies do allow observations of the belowground phenology with a higher resolution (i.e., up to once a day) but involve significant costs [43].

While knowing the fine-root lifespan, turnover, and phenology are important for climate modeling (especially for belowground processes and the parametrization of soil carbon fluxes) and timber production (due to the storage of carbohydrates in the roots), less is known about fine-roots dynamics than aboveground dynamics [19,23,44–48]. Three reasons may account for this knowledge gap. First, there are considerable difficulties in estimating the fine-root lifespan, turnover rate, and phenology, and none of the current measurement methods account for all these difficulties, which include: measuring a process of continuous growth and mortality that is not clearly defined, a relatively inaccessible study system, labor-intensive measurements, and soil disturbance. Second, as opposed to estimations of the leaf lifespan, our estimations of the fine-root lifespan, turnover rate, and phenology vary significantly among and within species and among and within years with very few trends [8,49]. In fact, estimations of the root lifespan can vary by more than fivefold among species, while estimations of the leaf lifespan are usually similar among species [50]. At first sight, the drivers controlling the fine-root lifespan, turnover, and phenology are unclear and seemingly unconnected to aboveground tree traits with few exceptions (e.g., the positive correlation between the fine-root diameter and wood production or fine-root biomass and aboveground biomass) [23,29,51]. Third, the distribution of root types and corresponding root characteristics can show significant spatial variability (even on a local scale), and the lifespan of fine-roots within the same diameter group may vary notably [1,8,9].

### 1.4. Research Questions and Hypotheses

This study expands our knowledge in the fine-root phenology of two late-successional (beech and oak) and one pioneer (birch) tree species. All three species have fine-roots that are colonized by ectomycorrhizae, known for their nitrogen, phosphorus, and water uptake ability [13,23,52]. Nevertheless, we expect differences in the fine-root lifespan, turnover rate, and phenology among species. For example, based on the difference in the resource strategy (i.e., acquisitive or conservative) and related root traits, the fine-root lifespan and turnover rate of birch is expected to be smaller and greater than those of beech and oak, respectively [40].

Because of the use of different methods and the measuring of roots at different depths, estimations of the fine-root lifespan and turnover rate vary widely in the literature [23,27,50,53–61]. Beech, oak, and birch were reported to have a mean fine-root lifespan of approximately 424 days, 358 days, and 935 days, respectively [50,62–65]. The fine-root turnover rate would then correspond for beech, oak, and birch to approximately 0.86 year$^{-1}$, 1.02 year$^{-1}$, and 0.39 year$^{-1}$, respectively. These estimates are opposite to the expectation that fine-roots of late-successional species live longer than those of pioneer species (with the literature reporting only a few estimations for oak and birch) [62–64,66,67]. Therefore, we aimed to clarify this confusion. However, one should note that these estimations of the fine-root lifespan and turnover rate in the literature might also have been affected by differences in the locations (e.g., the soil, the latitude, or the elevation) of the experiments and therefore might not solely represent species-specific differences.

In conclusion, this study addressed three questions: (I) How do the below- and aboveground phenology of deciduous trees relate to each other? (II) Are there differences in the fine-root lifespan, turnover rate, and phenology among beech, oak, and birch?

(III) Are there differences in the fine-root lifespan and turnover rate among roots with different diameters or those born in different seasons?

## 2. Methods

### 2.1. Description of the Sites

#### 2.1.1. Field Sites and Experimental Lay-Out

We observed the belowground phenology of the deciduous trees' fine-roots from April 2019 to March 2021. We selected dominant mature trees in the forests of the Klein Schietveld in Kapellen (KS; 51°21′ N, 4°37′ E) and the Park of Brasschaat (PB; 51°12′ N, 4°26′ E) near Antwerp (Belgium). The selection included four beeches (*Fagus sylvatica* L.) and four birches (*Betula pendula* Roth) in the KS and four beeches and four oaks (*Quercus robur* L.) in the PB. The trees were chosen because their aboveground wood and leaf phenology have been extensively monitored since 2017 in the framework of the LEAF-FALL project [68–73]. As such, concurrent measurements allowed the integration of results on both the below- and aboveground phenology in deciduous trees.

#### 2.1.2. Meteorological Conditions

The meteorological conditions at our study sites can be approximated using (long-term) values of the temperature, the precipitation, the number of rainy days, the relative humidity, the sunshine duration, and the global solar radiation measured at the Royal Meteorological Institute's (KMI) meteorological station in Ukkel (Table 1). These values were averaged for each season and compared against normal values for the reference periods of 1981–2010 and 1991–2020. Daily values of the temperature, the precipitation, the relative humidity, the vapor pressure deficit, and the volumetric water content are also presented here (Figure 1A–E). These were measured by the Flemish Institute for Nature and Forest (INBO) and the Integrated Carbon Observation System (ICOS) at the meteorological station of Brasschaat. When required, the data were gap-filled using measurements of the Royal Netherlands Meteorological Institute's (KNMI) meteorological station of Woensdrecht (The Netherlands). More information on the meteorological stations and the measurements methods can be found in the literature [71,74,75].

**Table 1.** Overview of the meteorological conditions perceived by the mature trees in the study region in 2019, 2020, and the winter of 2021. All data were measured by the meteorological station of the Royal Meteorological Institute (KMI) in Ukkel, Belgium [76–84]. Values within the five highest/lowest values since the reference periods 1981–2010 and 1991–2020 are marked by (+/-), while values within the three highest/lowest values are marked by (++/--). Record values are marked by (+++/---).

| | Normal (1981–2010) | | | | 2019 | | | | 2020 | | | | Normal (1991–2020) | 2021 |
| --- | --- | --- | --- | --- | --- | --- | --- | --- | --- | --- | --- | --- | --- | --- |
| | Winter | Spring | Summer | Autumn | Winter | Spring | Summer | Autumn | Winter | Spring | Summer | Autumn | Winter | Winter |
| **Average temperature (°C)** | 3.6 | 10.1 | 17.6 | 10.9 | 5.2 | 10.5 | 19.1 (++) | 11.3 | 6.3 (++) | 11.3 | 18.8 | 12.3 (+) | 4.1 | 4.7 |
| **Total precipitation (mm)** | 220.5 | 187.8 | 224.6 | 219.9 | 235.8 | 176.5 | 198.6 | 209.3 | 230.3 | 105.7 (-) | 168.2 | 219.2 | 228.6 | 264.1 |
| **Average number of rainy days** | 54.8 | 49 | 43.9 | 51 | 48 | 44 | 33 | 53 | 58 | 23 (---) | 46 | 43 | 55.2 | 54.8 |
| **Relative humidity (%)** | 84 | 74 | 73 | 82 | 84 (--) | 72 | 70 | 83 | 85 | 61 (--) | 66 (--) | 79 (--) | 84 | 84 |
| Sunshine duration (h:m) | 180:18 | 463:58 | 578:20 | 322:00 | 226:13 (+) | 489:42 | 714:38 (++) | 322:23 | 169:58 | 740:46 (+++) | 602:50 | 346:35 | 180:17 | 182:22 |
| Global solar radiation (kWh/m²) | 73.9 | 325 | 429.6 | 168.2 | 87.6 | 345.6 | 487.9 (+) | 178.4 | 73.3 | 61 (---) | 454.8 | 177 | 75.5 | 83.1 |
| Vapor pressure (hPa) | 6.9 | 9.2 | 14.5 | 11 | 6.9 | 9 | 15 | 11.2 | 8.2 (++) | 8.1 (--) | 13.9 | 11.2 | 7.1 | 7.4 |
| Air pressure (hPa) | 1017.3 | 1015.2 | 1016.2 | 1015.6 | 1016.4 | 1015.6 | 1015.4 | 1011 (--) | 1015.2 | 1017.8 | 1014.2 (--) | 1016.2 | 1017.1 | 1011.3 (-) |

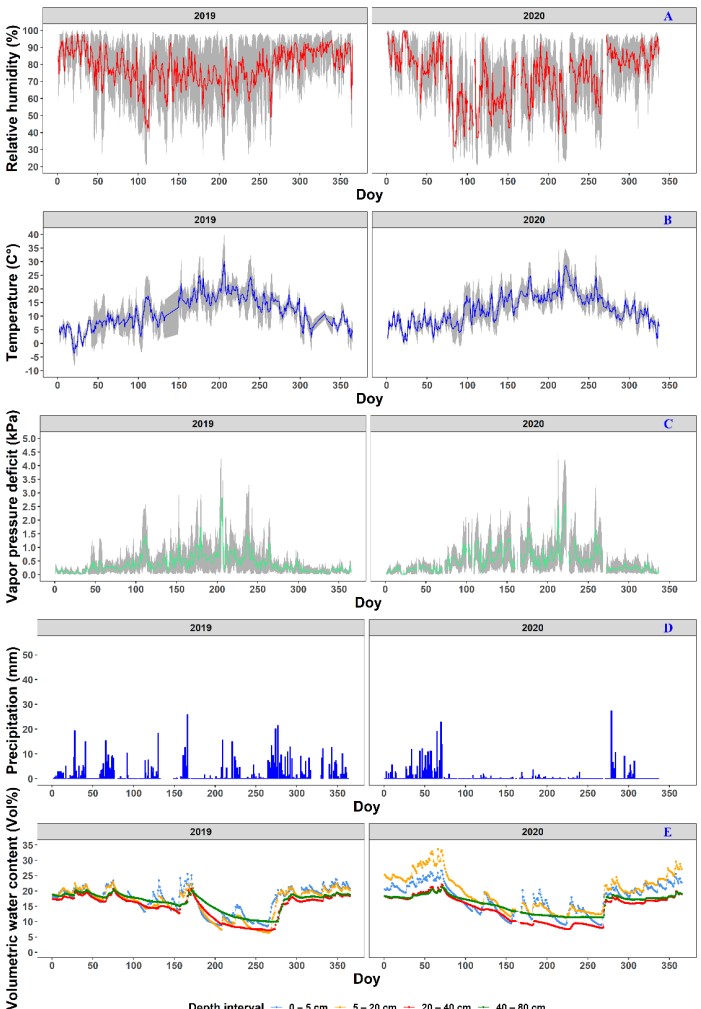

**Figure 1.** The meteorological conditions near the Klein Schietveld and Park of Brasschaat. The line plots represent the daily average relative humidity (%; **A**; red), the temperature (°C; **B**; blue), and the vapor pressure deficit (kPa; **C**; green), while the bar plots represent the daily precipitation (mm; **D**; light blue). The grey bands surrounding the line plots represent daily maxima and minima, respectively. The data were measured every half hour and were provided by the Flemish Institute for Nature and Forest (INBO), the Integrated Carbon Observation System (ICOS), and the Royal Dutch Meteorological Institute (KNMI). The vapor pressure deficit (kPa) was calculated using the formulas of Buck [85] using data of the relative humidity and air temperature between 7 a.m. and 7 p.m. The volumetric soil water content data (Vol%; **E**), provided through the courtesy of INBO, were measured every hour following De Vos [74]. The blue, orange, red and green dots represent the volumetric soil water content at depth intervals of 0–5 cm, 5–20 cm, 20–40 cm and 40–80 cm, respectively. This figure is partly adapted after Mariën et al. [71] and Mariën et al. [73].

To indicate the drought stress, we also present here the rainfall deficit for the hydrological years 2019 to 2021 (Figure 2). We first computed the potential evapotranspiration using the Bultot et al. [86] method and using data from the KMI's meteorological station in Ukkel (see Penman [87] and Baguis et al. [88] for details). By accumulating the daily potential evapotranspiration minus the daily sum of the precipitation, we were then able to compute the daily rainfall deficit per hydrological year (i.e., starting from a zero deficit around the first of April) and using continuous computation (i.e., when accounting for the hydrological fraction in the wet period that does not contribute to building up ground water reserves). The period from 1901 to 2000 was used as a long-term reference [71,73].

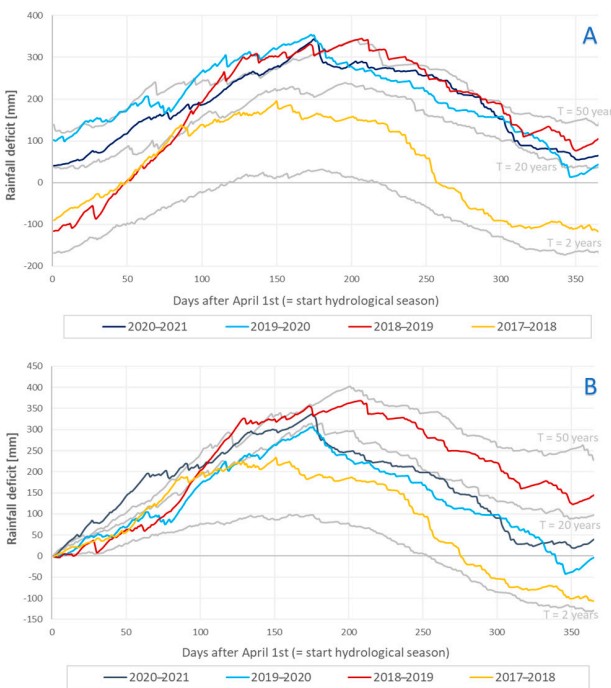

**Figure 2.** The rainfall deficit for the meteorological station of the Royal Meteorological Institute (KMI) in Ukkel, Belgium. Colored solid lines represent the rainfall deficit for the hydrological years in the period of 2019–2021, while grey solid lines represent the long-term reference statistics (computed for the 100-year period of 1901–2000). T stands for the return period, which represents the mean time between two successive exceedances of a given deficit value, and it was computed in an empirical way [89,90]. Panel **A** uses a continuous computation, while panel **B** starts from a zero deficit on the first of April (the start of the hydrological year). The colors represent the rainfall deficit in 2018–2019 (red), 2019–2020 (light blue), and 2020–2021 (purple). This figure was partly adapted after Mariën et al. [71] and Mariën et al. [73].

The climate at our study sites can be considered temperate maritime. Nevertheless, the weather during our study period was marked by a number of abnormal to exceptional events (Table 1). For example, in the summer of 2019, three heatwaves occurred, and the absolute maximum air temperature record for Belgium was broken. Consequently, the meteorological stations recorded an average air temperature and a total amount of sunshine falling in the three highest recorded values since 1981 [77]. Although, the autumn of 2019 could be considered normal, 2020 again recorded some abnormal to exceptional events [78,79]. For example, the winter of 2020 was abnormally warm with an average air temperature and vapor pressure falling in the three highest recorded values since 1981 [80]. This was followed by an extremely dry and sunny spring with, for Belgium, record-breaking values in the average duration of the sunshine, the global solar radiation the, relative humidity, the vapor pressure, and the number of rainy days [81]. The summer of 2020 was also dry and warm with an average relative humidity and air pressure falling in the three lowest recorded values since 1981 [82]. Finally, 2020 ended with a very warm autumn with an average relative humidity again falling in the three lowest recorded values since 1981 [83]. The winter of 2021 showed meteorological values that can be considered normal given the reference period f 1991–2020 [84]. Data on the volumetric water content for 2019 and 2020 showed, especially in summer, relatively low values. Since 2017, the rainfall deficit also showed an increasing trend. In the hydrological years of 2019–2020 and 2020–2021, the rainfall deficit even reached values similar to the hypothetical values forecasted for a return period of 100 years [73].

### 2.1.3. Soil Conditions

The soils were classified according to the Belgian soil classification system by texture, drainage status, and profile development [91–93]. At the KS, the soils of the beeches and birches were classified as Zdg and X (or ZAg), respectively. At the PB, the soils of the beeches and oaks were classified as Zcm and W-Zdg, respectively. Van Ranst and Sys [92] defined these soil types as follows: "The Zdg and W-Zdg soils are moderately wet and sandy with a clear humus or iron B horizon. The horizons in both soil types show tracks of oxidation or reduction. However, the iron B horizon is more clustered in W-Zdg soils than Zdg soils. While Zdg and W-Zdg soils have a suitable water management in summer, they are exposed to high water levels during winter. The X (or ZAg) soils consist of an approximately 40 cm thick layer of dune sand followed by black layers of decomposed plant remains. Finally, the Zcm soils are moderately dry and sandy with a deep anthropogenic A horizon" (p. 100, 198, and 201). Compared to the World Reference Base for Soil Resources (WRB), the Zdg, W-Zdg, Zcm, and X (or ZAg) soils would roughly coincide with aric-albic/arenic and endogleyic podzols, plaggic anthrosols and albic/aric-albic/brunic, and endogleyic podzols/arenosols, respectively [91,94,95]. Research by De Vos [75] has shown that the soil temperature and moisture regimes in the soil near the meteorological station of Brasschaat were mesic (i.e., "between 8 °C and 15 °C, and with a difference of 5 °C between the mean summer and winter temperature at 50 cm below the surface"; Chesworth et al. [96]; p. 671) and udic (i.e., "not dry in the relevant soil sections for more than 90 cumulative days in normal years or 60 consecutive days following summer solstice when the soil temperature at 50 cm below the surface is above 5 °C"; Chesworth [97]; p. 485).

### 2.2. Observing and Measuring Fine-Roots

In May 2018, we installed three transparent tubes of cellulose acetate butyrate near each selected tree. The tubes, with a length and a diameter of 60 cm and 5.2 cm, respectively, were installed in one line at distances of one, one-and-a-half, and two meters from the trees at an angle of $\pm30°$ to the soil surface. As such, the tubes reached a vertical viewing depth of $\pm35$ cm [4]. Because the installation of these tubes disturbed the soil and fine-root production, we waited until April 2019 for the first acquisition of images [98,99]. During the stabilization period, four tubes were damaged. New tubes were installed allowing less time for the soil and fine-root production near these tubes to reach equilibrium again. During the period of May 2018 to March 2021, all tubes were sealed using a rubber cork lid and covered by an empty plastic bottle cut in half. Nevertheless, some tubes proved to be not completely waterproof. Therefore, before each sampling session, we removed moisture in the tubes using a sponge and a hand siphon pump.

From April 2019 to March 2021, we collected images of the fine-roots using the manual BTC-2 minirhizotron camera and the associated I-CAP software (Bartz Technology Corp., Carpinteria, CA, USA). Thanks to a small hole in the upper side of the tubes and the smucker handle of the minirhizotron, a stable and consequent positioning of the minirhizotron on the tubes was possible. We were able to acquire for each month and from each tube a series of 22 images taken from the same downwards viewpoint, allowing repetitive and non-destructive measurements of the same fine-roots over time. Each image represented a soil area with a width of 13 mm and a length of 9 mm. Finally, all images were analyzed using the software WinRHIZO Tron MF 2015b (Regent Instruments Inc., Quebec, Canada), resulting in temporal data on the number, birth, death, length, surface area, average diameter, and volume of the fine-roots. We note here that the dates of the birth and the death of the fine-roots were the dates of the sampling session at which the fine-roots were first or last observed, respectively.

*2.3. Aboveground Phenological Data*

For a descriptive comparison against the phenology of the fine-roots, we also collected data on the timing of the bud burst in spring and the decline of the chlorophyll content index (CCI) in autumn from which the timing of the leaf senescence can be derived [70].

For both 2019 and 2020, the bud burst was observed weekly or twice a week from the beginning of March until the end of May using a spotting scope (Swarovski Optik, Tyrol, Austria). At each session, we assessed ten buds of each tree following the scale adapted from Vitasse et al. [99] and Marchand et al. [72]; 0: dormant bud, 1: swelling bud, 2: bud-burst, 3: emerging leaf, and 4: one leaf completely detached from the bud. The data were subsequently scaled between 0 and 1 and interpolated with a loess regression. The timing of the trees' bud-burst was defined as the moment when the loess regression model reached a value of 0.5.

Every two weeks in 2019 and every week in 2020, from mid-July to late November, tree-climbers collected five sun-leaves and five shade-leaves from each tree. The CCI, a proxy for the chlorophyll concentration, of these leaves was immediately measured after harvest using a chlorophyll content meter (CCM-200 plus, Opti-Sciences Inc., Hudson, NH, USA). Subsequently, the trend in the CCI was modeled following Mariën et al. [74] using generalized additive models for location, scale, and shape [100–102]. A local minimum in the second derivative of these trends was then used to derive the phenological transition date. This date represents the date when the decline in the CCI accelerates most rapidly and is a proxy for the onset of the relevant processes occurring during the re-organization phase in the senescence process. In other words, it indicates the timing of the phase when a rapid decrease in the photosynthetic activity and chlorophyll concentrations is initiated, usually as a consequence of increases in reactive oxygen species or abscisic acid concentrations and decreases in cytokine concentrations [103]. The re-organization phase is also the phase that allows remobilization of nutrients from the leaves to other crucial plant organs, and it is the phase when the first signs of ultrastructural change in the leaf cell becomes visible [104]. After resampling, the likelihood of a leaf to start senescing can be approximately represented by the resulting distribution of the phenological transition dates. Details on the methods for collecting and modeling the CCI and the phenological transition date are covered extensively in Mariën et al. [71] and Mariën et al. [72].

*2.4. Statistical Analyses*

Unless otherwise specified, all analyses were done using R v3.6.3. and the packages R/dplyr, R/ggplot2, R/cowplot, and R/viridis [105–109]. Model assumptions were tested following Zuur et al. [110].

2.4.1. Detecting Trends in the Fine-Root Phenology Using Generalized Additive Mixed Models

Due to the relatively long period between sampling sessions, we seldom accurately observed elongation within one single root. Therefore, only two variables could be used to present the trend in the fine-root phenology per species: the new root count per tube (i.e., the root initiation rate or difference in the number of new roots between two consecutive sessions) and the cumulative root surface per tube (i.e., the sum of each roots' surface area per tube; in mm$^2$). As we did not account for the root ontology or order, each root count here actually reflects an observed root tip. Each root was classified by its diameter d (d < 2 mm, 1 mm < d < 2 mm, 0.5 mm < d < 1 mm, and d < 0.5 mm) and season of birth (spring, summer, autumn, and winter). Unless otherwise specified, the uncertainty presented here is the variability among the twelve to twenty-four tubes per species.

The trend in mean new root count and cumulative root surface was assessed using generalized additive models (GAMs) built using the *gam* function in R/mgcv [111,112]. Although GAMs are not as flexible as generalized additive models for location, scale, and shape (GAMLSS), we used GAMs here to ease the plotting, diagnostic testing, and interpretation of the model results using the *summary*, *gam.check*, *acf*, *pacf*, and *plot* function

in R/mgcv and the *confint* function in R/gratia [113–115]. The downside of GAMs in comparison to more elaborate models like GAMLSS is that information on the trend in the variance, skewness, and kurtosis is lost [116]. Therefore, we comment below on potential model improvements where interesting.

To model the new root count and the cumulative root surface as functions of their covariates, two negative binomial GAMs with the default logarithmic link function and two GAMs with the gamma distribution and the default inverse link functions were made, respectively. The GAMs with the negative binomial and gamma distribution families were chosen because they showed the lowest Akaike Information Criterion (AIC). Two GAMs (one with a negative binomial and one with a gamma distribution) were used to model the cumulative root surface and new root count of the oak trees in the PB and the birch and beech trees in the KS (Models 1–2). The fixed covariates of these GAMs were the *day* (counted from the first of January; continuous) and *species* (categorical with three levels). These two covariates were modeled as a factor-smooth interaction term and smoothed using P-splines [117,118]. Due to the high number of zeros in the data, two other GAMs (again one with a negative binomial and one with a gamma distribution) were used to model the cumulative root surface and the new root count of the beech trees in the PB (Models 3–4). The fixed covariate of these GAMs was the *day*, which was also smoothed using P-splines. In all GAMs, the *individual tree* (categorical with 16 levels) was modeled as a random intercept to account for the dependency among observations of the same individual tree. The restricted maximum likelihood (REML) argument was specified as the smoothness selection method to avoid overfitting [112,119].

$$
\begin{gathered}
Y_{ij} \sim NB(\mu_{ij,}, k) \\
g(\mathbb{E}(Y_{ij})) = g(\mu_{ij}) \\
g(\mu_{ij}) = Species_{ij} + f(Day_{ij}, Species_{ij}) + Individual\ tree_i
\end{gathered}
\tag{1}
$$

where $\mu_{ij}$ is the conditional mean; k is the dispersion parameter; g is the logit link function; $\mu_{ij}$ is the conditional mean; $\mathbb{E}(Y_{ij})$ are the expected values of the response variable $Y_{ij}$; $f(x_{ij})$ is the smooth function of the covariate $x_{ij}$; $Y_{ij}$ is the *j*th observation of the response variable (i.e., the new root count) in individual tree *i*, and *i* = 1, . . . , 16; and the individual tree$_i$ is the random intercept.

$$
\begin{gathered}
Y_{ij} \sim Gamma(\mu_{ij,}, k) \\
g(\mathbb{E}(Y_{ij})) = g(\mu_{ij}) \\
g(\mu_{ij}) = Species_{ij} + f(Day_{ij}, Species_{ij}) + Individual\ tree_i
\end{gathered}
\tag{2}
$$

where $\mu_{ij}$ is the conditional mean; k is the dispersion parameter; g is the logit link function; $\mu_{ij}$ is the conditional mean; $\mathbb{E}(Y_{ij})$ are the expected values of the response variable $Y_{ij}$; $f(x_{ij})$ is the smooth function of the covariate $x_{ij}$; $Y_{ij}$ is the *j*th observation of the response variable (i.e., the cumulative root surface) in individual tree *i*, and *i* = 1, . . . , 16; and the individual tree$_i$ is the random intercept.

$$
\begin{gathered}
Y_{ij} \sim NB(\mu_{ij,}, k) \\
g(\mathbb{E}(Y_{ij})) = g(\mu_{ij}) \\
g(\mu_{ij}) = f(Day_{ij}) + Individual\ tree_i
\end{gathered}
\tag{3}
$$

where $\mu_{ij}$ is the conditional mean; k is the dispersion parameter; g is the logit link function; $\mu_{ij}$ is the conditional mean; $\mathbb{E}(Y_{ij})$ are the expected values of the response variable $Y_{ij}$; $f(x_{ij})$ is the smooth function of the covariate $x_{ij}$; $Y_{ij}$ is the *j*th observation of the response variable (i.e., the new root count) in individual tree *i*, and *i* = 1, . . . , 16; and the individual tree$_i$ is the random intercept.

$$
\begin{gathered}
Y_{ij} \sim Gamma(\mu_{ij,}, k) \\
g(\mathbb{E}(Y_{ij})) = g(\mu_{ij}) \\
g(\mu_{ij}) = f(Day_{ij}) + Individual\ tree_i
\end{gathered}
\tag{4}
$$

where $\mu_{ij}$ is the conditional mean; k is the dispersion parameter; g is the logit link function; $\mu_{ij}$ is the conditional mean; $\mathbb{E}(Y_{ij})$ are the expected values of the response variable $Y_{ij}$; $f(x_{ij})$ is the smooth function of the covariate $x_{ij}$; $Y_{ij}$ is the *j*th observation of the response variable (i.e., the cumulative root surface) in individual tree *i*, and *i* = 1, . . . , 16; and the individual tree$_i$ is the random intercept.

### 2.4.2. Estimating the Fine-Root Lifespan and Turnover Rate Using Kaplan–Meier Survival Analyses

The fine-root lifespan and its inverse, the turnover rate, were estimated using Kaplan–Meier survival analyses [120]. This type of non-parametric analysis allows to accurately model right-censored data (i.e., in this case, roots of which the time of death could not be determined because they were alive at the end or because they accidentally disappeared before the end of the study period) and to make estimations of the survival probability divided by cohort. The Kaplan–Meier survival analyses were built and analyzed using the *surv*, *survfit*, *summary*, and *print* functions from the R/survival package [121,122]. The models were graphically explored using the *ggsurvplot* function from the package R/survminer [123].

One Kaplan–Meier survival analysis was performed on the overall data of the fine-roots (i.e., the roots with a diameter below 2 mm) using the *species* as a covariate. Two other Kaplan–Meier survival analyses, one with the *root diameter class* as a covariate and one with the *season of root birth* as a covariate, were performed on the fine-root data subdivided per species. Differences between the survival curves could be tested using the non-parametric log-rank test implemented by the *survdiff* function in R/survival [124,125]. Subsequently, a post-hoc multiple comparison analysis with Benjamini–Hochberg correction could be implemented using the *pairwise_survdiff* function in R/survminer. The resulting median survival time and its confidence intervals are estimations of the fine-root lifespan.

## 3. Results

### 3.1. Trends in the Cumulative Root Surface and Root Count

All trends of the mean cumulative root surface and new root count were significant with *p*-values below 0.05, except for the mean cumulative root surface of birch (Figure 3, Figure 4A,B; Table 2). However, these trends did not show clear yearly patterns and were inconsistent among years.

In the beech trees of the KS, the mean cumulative root surface increased sharply until a plateau was reached around late summer 2019. The mean cumulative root surface then still increased until the beginning of spring 2020. Afterwards, the mean cumulative root surface steeply decreased. The mean new root count per tube near the beech trees of the KS slowly decreased from the start of the measurements until approximately December 2019. However, unlike the trends resulting from the GAMs, the data of 2019 show two peaks in the new root count per tube: one around July and one around October. After December 2019, the beech trees of the KS seemingly produced almost no new roots. The mean cumulative root surface in the beech trees of the PB showed much less activity than the beech trees of the KS, with only a small peak around the beginning of September 2019 and another minor increase in September 2020. Afterwards, the mean cumulative root surface quickly decreased again to constant and very low levels. A relatively similar pattern was observed in the mean new root count per tube in the beech trees of the PB. A small peak of new roots was observed around the beginning of September 2019. This was followed by a steep decrease, reaching, again, constant and very low levels.

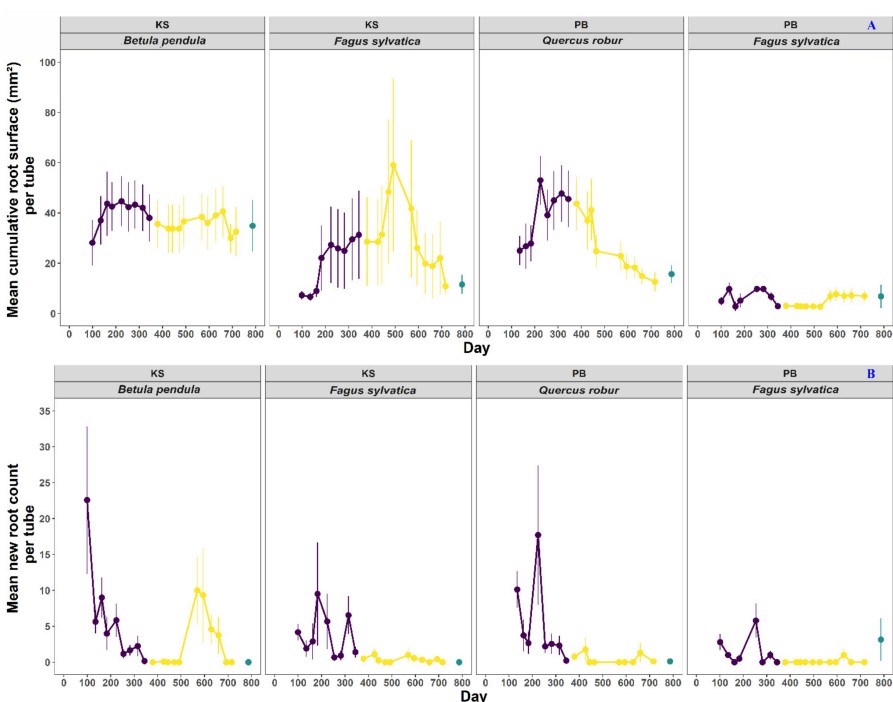

**Figure 3.** The cumulative surface area (**A**) and new root count (**B**) per tube of the mature *Fagus sylvatica* ($n_{KS}$ = 12; $n_{PB}$ = 12), *Quercus robur* (*n* = 12), and *Betula pendula* (*n* = 12) trees at the Klein Schietveld and Park of Brasschaat. Dots and error bars represent the mean cumulative root surface and new root count with standard errors, respectively. The purple, yellow, and green colors represent the sampling years 2019, 2020, and 2021, respectively. Days were counted from the first of January 2019.

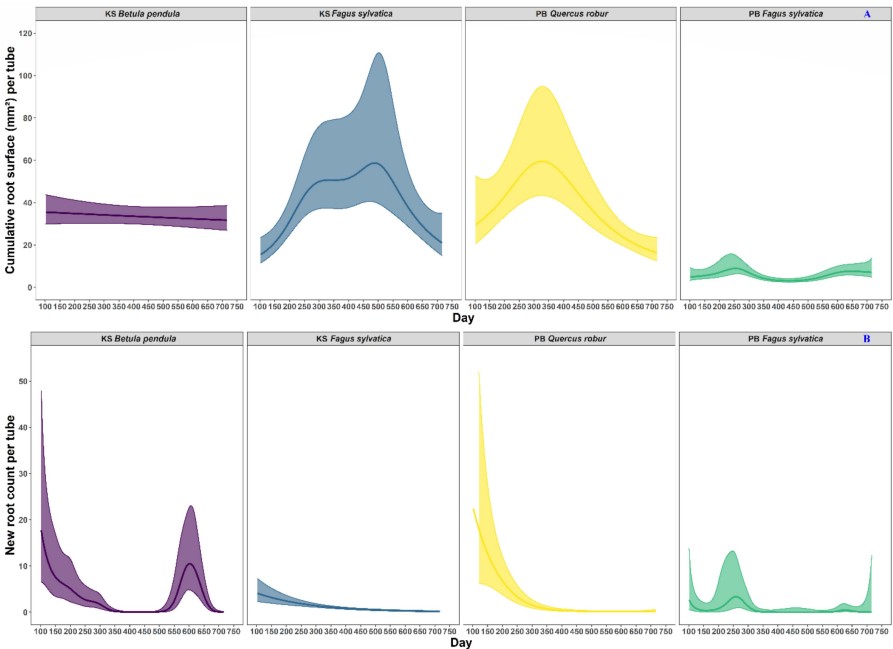

**Figure 4.** The generalized additive model fits for the cumulative surface area (**A**) and new root count (**B**) per tube of the mature *Fagus sylvatica* ($n_{KS}$ = 12; $n_{PB}$ = 12), *Quercus robur* (*n* = 12), and *Betula pendula* (*n* = 12) trees at the Klein Schietveld and Park of Brasschaat. The colored solid lines represent smooth terms of the mean, while the light and dark colored shaded bands around the smooth terms represent approximate 95% simultaneous confidence intervals and 95% pointwise confidence intervals, respectively. The green, yellow, and purple colors represent the species *Fagus sylvatica*, *Quercus robur*, and *Betula pendula*, respectively. Days were counted from the first of January 2019.

**Table 2.** Adjusted $R^2$, effective degrees of freedom (edf), and F-test values of the GAM smooth terms (*Day* counted from the first of January). $\mathbb{E}(y_i)$ are the expected values of the response variable $y_i$; $f(x_i)$ is the smooth function of the covariate $x_i$; $\beta_i$ is the intercept of the covariate $x_i$; $\zeta$ is the random effect; and $\varepsilon_i$ are the errors. All smooth functions were fitted using P-splines.

| $Y_i$ | Model Equation | Family Distribution | Link Function | Adjusted $R^2$ | Site | Smooth Term | Species | Edf | F or Chi.sq | *p*-Value |
|---|---|---|---|---|---|---|---|---|---|---|
| New root count | $g(\mathbb{E}(y_i)) = f_1(Day_i) + \zeta_{ID} + \varepsilon_i$ | Negative binomial | Logarithmic | 0.28 | PB | Day | *Fagus sylvatica* L. | 6 | 18 | <0.05 |
| New root count | $g(\mathbb{E}(y_i)) = \beta_1 Species_i + f_1 Species_i(Day_i) + \zeta_{ID} + \varepsilon_i$ | Negative binomial | Logarithmic | 0.11 | KS | Day | *Fagus sylvatica* L. | 1 | 30 | <0.001 |
| | | | | | PB | | *Quercus robur* L. | 2.6 | 59 | <0.001 |
| | | | | | KS | | *Betula pendula* Roth | 7.5 | 89 | <0.001 |
| Cumulative root surface | $g(\mathbb{E}(y_i)) = f_1(Day_i) + \zeta_{ID} + \varepsilon_i$ | Gamma | Inverse | 0.2 | PB | Day | *Fagus sylvatica* L. | 5.3 | 3.6 | <0.01 |
| Cumulative root surface | $g(\mathbb{E}(y_i)) = \beta_1 Species_i + f_1 Species_i(Day_i) + \zeta_{ID} + \varepsilon_i$ | Gamma | Inverse | 0.25 | KS | Day | *Fagus sylvatica* L. | 4 | 4.2 | <0.001 |
| | | | | | PB | | *Quercus robur* L. | 2.8 | 6.5 | <0.001 |
| | | | | | KS | | *Betula pendula* Roth | 1 | 0.6 | ns |

In oak, the mean cumulative root surface increased until a plateau was reached around the beginning of summer 2019. This plateau remained until the end of autumn 2019 and was followed by a relatively fast subsequent decline in the early spring of 2020. The pattern in the mean new root count per tube in oak was similar to the pattern found in the beech trees of the KS, with very few roots produced in 2020. However, the initial number of new root counts in oak was approximately fivefold higher than those of beech. Birch presented a stable cumulative root surface for the whole measurement period. Like beech and oak, the mean new root count per tube in birch decreased from the beginning of the measurements until approximately December 2019. However, unlike in beech and oak, the mean new root count per tube in birch increased again in the spring of 2020. In late August 2020, a peak was reached in the mean new root count per tube in birch. This peak was followed by a decrease until the beginning of December 2020 when seemingly no new fine-roots were produced.

### 3.2. Linking the Above- and Belowground Phenology of Fagus sylvatica L., Quercus robur L., and Betula pendula Roth

In 2019 and 2020, the bud burst occurred during the first week of April in birch and during the second to third week in beech and oak (Figure 5A). The mean phenological transition date suggests that the onset of the re-organization phase of leaf senescence in 2019 took place somewhere around the second week of September in oak, the second week of October in birch and the fourth week of October in beech (Figure 5B,C). In 2020, the mean phenological transition date suggests a similar timing for beech but a slightly earlier and later onset of the re-organization phase of senescence for birch and oak, respectively. This would be around the fourth week of September for birch and the fourth week of October for oak. However, one should note here that the distribution of the phenological transition dates of oak in 2019 and birch in 2020 is strongly bimodal. Therefore, the mean phenological transition date is not likely the most representative metric for the onset of the re-organization phase of senescence here. A more nuanced view suggests that beech indeed has a stable timing in the onset of its re-organization phase of senescence around the third week of October [73]. On the other hand, oak has an onset of its re-organization phase of leaf senescence that is mainly around the fourth week of October. However, occasionally, oak can have an antecedent wave in its re-organization phase of senescence that starts earlier (e.g., as in 2019 around the fourth week of July) and is possibly related to drought [126,127]. Birch, which is known for its continuous leaf flushing and nondeterministic growth pattern, shows various peaks of its re-organization phase of leaf senescence.

In 2019 and 2020, birch showed a cumulative root surface that was constant during autumn (i.e., $\pm$40 mm$^2$ of fine root surface per tube). Nevertheless, new roots were still being produced during the autumn of 2019 and, in particular, of 2020. The cumulative root surface of beech and oak remained relatively high (i.e., $\pm$30 to 50 mm$^2$ of fine root surface per tube for beech and oak, respectively) and stable during the autumn of 2019 but declined during the autumn of 2020. For beech, peaks of new roots count (i.e., $\pm$10 new roots per tube) were observed in 2019 in October (at the KS) or September (at the PB) concurrently or earlier than the senescence phase. However, almost no new roots were produced in beech during the autumn of 2020 and in oak during both autumns. The cumulative root surface of birch was also constant during the bud burst in 2019 and 2020. However, the cumulative root surface of beech did seem to increase during the bud burst in 2019 and 2020. In oak, the cumulative root surface increased and decreased during the bud burst of 2019 and 2020, respectively. Only during the bud burst of 2019 did beech and oak produce new roots.

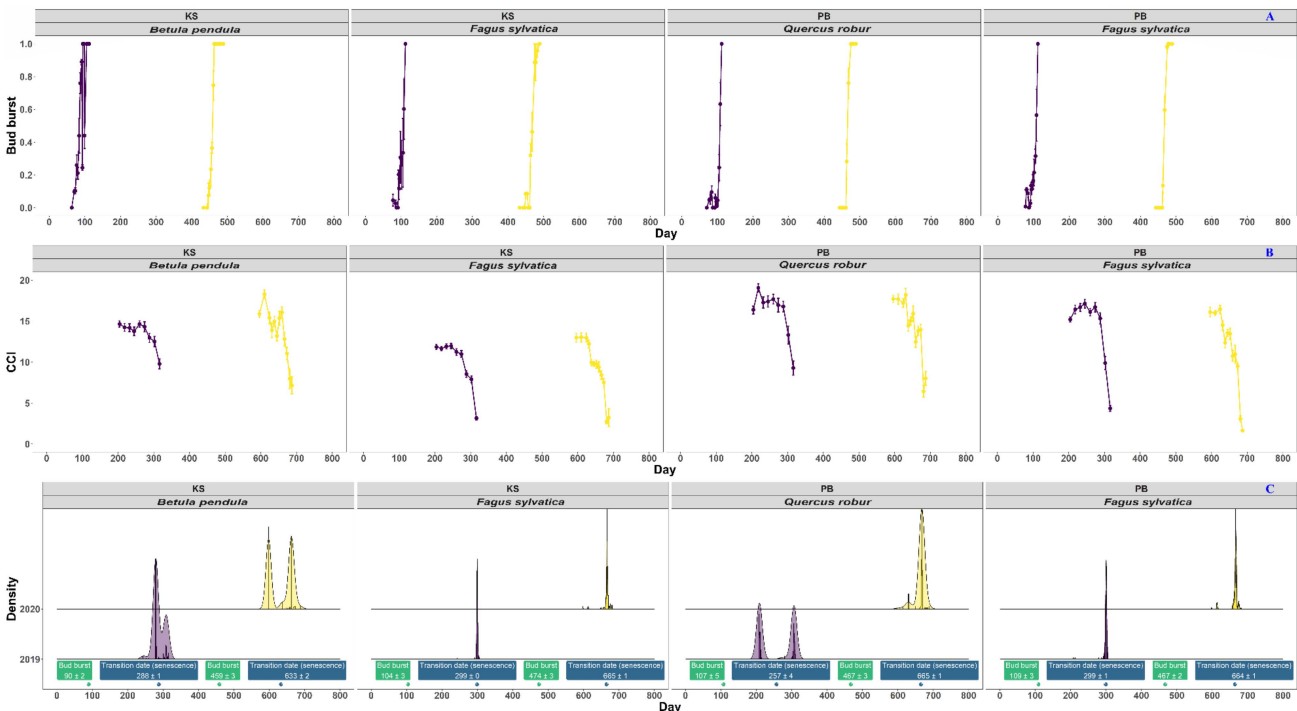

**Figure 5.** The bud burst (**A**), chlorophyll content index (CCI; **B**), and distribution of the phenological transition dates (a proxy for the onset of senescence; **C**) of the mature *Fagus sylvatica* ($n_{KS}$ = 4; $n_{PB}$ = 4), *Quercus robur* (*n* = 4), and *Betula pendula* (*n* = 4) trees at the Klein Schietveld and Park of Brasschaat. Dots and error bars represent the mean bud burst phase and CCI with standard errors, respectively. The purple and yellow colors represent the sampling years 2019 and 2020, respectively. The four bottom plots show the relative ecological timing of the mean bud burst and the mean phenological transition date per site and species. Days were counted from the first of January 2019.

### 3.3. Fine-Root Lifespan and Turnover Rate

Estimations of the fine-root lifespan, the turnover rate, and their confidence intervals are reported in Table 3, while a comparison of the statistical significance among the Kaplan–Meier survival curves is reported in Table 4. The estimations on the median lifespan and the turnover rate in the fine-roots per species ranged from 440 days and 0.83 year$^{-1}$ in birch to 470 days and 0.78 year$^{-1}$ in beech, respectively (Table 3). The Kaplan–Meier survival curves of the fine-roots also differed significantly among the species (Table 4). However, within one species, there were also significant differences in the estimations of the median lifespan and the turnover rate between roots with a different diameter class or season of birth. For example, for oak, estimations for the median lifespan and the turnover rate per root diameter class ranged from 241 days and 1.51 year$^{-1}$ for the fine-roots below 0.5 mm to 526 days and 0.69 year$^{-1}$ for the fine-roots above 2 mm, respectively. Likewise, estimations of the median lifespan and root turnover rate per season of root birth in birch ranged from 379 days and 0.96 year$^{-1}$ for the fine-roots born in autumn to 626 days and 0.58 year$^{-1}$ for the fine-roots born in winter (Table 3). For beech and oak, the variability in the fine-root lifespan across diameter classes was larger than the variability in the fine-root lifespan across the season of root birth. For birch, the trends were opposite.

**Table 3.** Estimations of the mean and median fine-root lifespan (days) and fine-root turnover rate (year$^{-1}$) using Kaplan–Meier survival analyses. N represents the number of observed fine-roots, while events represent the number of fine-roots, of which death was observed.

| | | | Survival Curve Per Species (with Root Diameter < 2 mm) | | | | | | | |
|---|---|---|---|---|---|---|---|---|---|---|
| Species | n | Events | Mean Lifespan (Days) | | Median Lifespan (Days) | | | Root Turnover Rate (Year$^{-1}$) | | |
| | | | Mean | Mean SE | Median | Lower 95% CI | Upper 95% CI | Rate | Lower 95% CI | Upper 95% CI |
| *Fagus sylvatica* | 4272 | 4165 | 412 | 3 | 470 | 463 | 470 | 0.78 | 0.79 | 0.78 |
| *Quercus robur* | 2511 | 2434 | 442 | 4 | 460 | 447 | 468 | 0.79 | 0.82 | 0.78 |
| *Betula pendula* | 6529 | 6228 | 426 | 3 | 440 | 408 | 470 | 0.83 | 0.89 | 0.78 |
| | | | Survival Curve for *Fagus sylvatica* Per Root Diameter Class | | | | | | | |
| Root Diameter Class | n | Events | Mean lifespan (Days) | | Median lifespan (Days) | | | Root turnover rate (Year$^{-1}$) | | |
| | | | Mean | Mean SE | Median | Lower 95% CI | Upper 95% CI | Rate | Lower 95% CI | Upper 95% CI |
| Root Diameter > 2 mm | 2321 | 2260 | 418 | 4 | 444 | 435 | 447 | 0.82 | 0.84 | 0.82 |
| 1 mm < Root Diameter > 2 mm | 3470 | 3371 | 409 | 3 | 470 | 447 | 470 | 0.78 | 0.82 | 0.78 |
| 0.5 mm < Root Diameter > 1 mm | 516 | 513 | 438 | 8 | 470 | 435 | 470 | 0.78 | 0.84 | 0.78 |
| Root Diameter < 0.5 mm | 286 | 281 | 399 | 12 | 511 | 310 | 511 | 0.71 | 1.18 | 0.71 |
| | | | Survival Curve for *Fagus sylvatica* Per Season of Root Birth (with Root Diameter < 2 mm) | | | | | | | |
| Season of Root Birth | n | Events | Mean Lifespan (Days) | | Median Lifespan (Days) | | | Root Turnover Rate (Year$^{-1}$) | | |
| | | | Mean | Mean SE | Median | Lower 95% CI | Upper 95% CI | Rate | Lower 95% CI | Upper 95% CI |
| Autumn | 1497 | 1497 | 399 | 5 | 470 | 447 | 470 | 0.78 | 0.82 | 0.78 |
| Spring | 884 | 884 | 406 | 6 | 441 | 379 | 470 | 0.83 | 0.96 | 0.78 |
| Summer | 1271 | 1271 | 428 | 5 | 470 | 470 | 495 | 0.78 | 0.78 | 0.74 |
| Winter | 620 | 513 | 417 | 7 | 447 | 406 | 470 | 0.82 | 0.90 | 0.78 |
| | | | Survival Curve for *Quercus robur* Per Root Diameter Class | | | | | | | |
| Root Diameter Class | n | Events | Mean Lifespan (Days) | | Median Lifespan (Days) | | | Root Turnover Rate (Year$^{-1}$) | | |
| | | | Mean | Mean SE | Median | Lower 95% CI | Upper 95% CI | Rate | Lower 95% CI | Upper 95% CI |
| Root diameter > 2 mm | 1468 | 1408 | 471 | 5 | 526 | 506 | 555 | 0.69 | 0.72 | 0.66 |
| 1 mm < root diameter > 2 mm | 2001 | 1935 | 446 | 4 | 460 | 460 | 468 | 0.79 | 0.79 | 0.78 |
| 0.5 mm < root diameter > 1 mm | 299 | 290 | 502 | 8 | 499 | 460 | 534 | 0.73 | 0.79 | 0.68 |
| Root diameter < 0.5 mm | 211 | 209 | 308 | 14 | 241 | 220 | 376 | 1.51 | 1.66 | 0.97 |
| | | | Survival Curve for *Quercus robur* Per Season of Root Birth (with Root Diameter < 2 mm) | | | | | | | |
| Season of Root Birth | n | Events | Mean Lifespan (Days) | | Median Lifespan (Days) | | | Root Turnover Rate (Year$^{-1}$) | | |
| | | | Mean | Mean SE | Median | Lower 95% CI | Upper 95% CI | Rate | Lower 95% CI | Upper 95% CI |
| Autumn | 912 | 912 | 446 | 6 | 460 | 447 | 473 | 0.79 | 0.82 | 0.77 |
| Spring | 401 | 432 | 432 | 10 | 447 | 437 | 468 | 0.82 | 0.84 | 0.78 |
| Summer | 757 | 757 | 444 | 7 | 460 | 460 | 468 | 0.79 | 0.79 | 0.78 |
| Winter | 441 | 364 | 437 | 10 | 447 | 437 | 468 | 0.82 | 0.84 | 0.78 |

**Table 3.** *Cont.*

| | | | Survival Curve for *Betula pendula* Per Root Diameter Class | | | | | | | |
|---|---|---|---|---|---|---|---|---|---|---|
| **Root Diameter Class** | **n** | **Events** | **Mean Lifespan (Days)** | | **Median Lifespan (Days)** | | | **Root Turnover Rate (Year⁻¹)** | | |
| | | | **Mean** | **Mean SE** | **Median** | **Lower 95% CI** | **Upper 95% CI** | **Rate** | **Lower 95% CI** | **Upper 95% CI** |
| Root diameter > 2 mm | 3765 | 3620 | 425 | 4 | 407 | 394 | 433 | 0.90 | 0.93 | 0.84 |
| 1 mm < root diameter > 2 mm | 5509 | 5260 | 422 | 3 | 440 | 394 | 470 | 0.83 | 0.93 | 0.78 |
| 0.5 mm < root diameter > 1 mm | 776 | 736 | 465 | 8 | 562 | 435 | 595 | 0.65 | 0.84 | 0.61 |
| Root diameter < 0.5 mm | 244 | 232 | 386 | 15 | 394 | 379 | 470 | 0.93 | 0.96 | 0.78 |
| | | | Survival Curve for *Betula pendula* Per Season of Root Birth (with Root Diameter < 2 mm) | | | | | | | |
| **Season of Root Birth** | **n** | **Events** | **Mean Lifespan (Days)** | | **Median Lifespan (Days)** | | | **Root Turnover Rate (Year⁻¹)** | | |
| | | | **Mean** | **Mean SE** | **Median** | **Lower 95% CI** | **Upper 95% CI** | **Rate** | **Lower 95% CI** | **Upper 95% CI** |
| **Autumn** | 2240 | 2240 | 376 | 5 | 379 | 358 | 379 | 0.96 | 1.02 | 0.96 |
| **Spring** | 1642 | 1642 | 479 | 5 | 511 | 471 | 559 | 0.71 | 0.77 | 0.65 |
| **Summer** | 1832 | 1832 | 387 | 5 | 387 | 371 | 394 | 0.94 | 0.98 | 0.93 |
| **Winter** | 815 | 514 | 544 | 6 | 626 | 595 | 653 | 0.58 | 0.61 | 0.56 |

**Table 4.** Comparison of the Kaplan–Meier survival curves using log-rank tests and post-hoc multiple comparison analyses.

| | Survival Curve Per Species (with Root Diameter < 2 mm) | | |
|---|---|---|---|
| | **Post-Hoc Analysis** | | **Log-Rank Test** |
| | *p* Value | | *p* Value |
| **Species** | *Fagus sylvatica* | *Quercus robur* | |
| *Fagus sylvatica* | - | 0.001 | <0.001 |
| *Betula pendula* | 0.001 | 0.001 | |
| | Survival Curve for *Fagus sylvatica* Per Root Diameter Class | | |
| | **Post-Hoc Analysis** | | **Log-Rank Test** |
| | *p* Value | | *p* Value |
| **Root Diameter Class** | **Root Diameter < 0.5 mm** | **Root Diameter > 2 mm** | **0.5 mm < Root Diameter > 1 mm** |
| Root Diameter > 2 mm | ns | - | - | <0.001 |
| 0.5 mm < Root Diameter > 1 mm | ns | ns | - |
| 1 mm < Root Diameter > 2 mm | <0.05 | <0.001 | <0.001 |
| | Survival Curve for *Fagus sylvatica* Per Season of Root Birth (with Root Diameter < 2 mm) | | |
| | **Post-Hoc Analysis** | | **Log-Rank Test** |
| | *p* Value | | *p* Value |
| **Season of root birth** | **Autumn** | **Spring** | **Summer** |
| Spring | ns | - | - | <0.001 |
| Summer | <0.001 | ns | - |
| Winter | <0.001 | <0.01 | ns |
| | Survival Curve for *Quercus robur* Per Root Diameter Class | | |
| | **Post-Hoc Analysis** | | **Log-Rank Test** |
| | *p* Value | | *p* Value |
| **Root Diameter Class** | **Root Diameter < 0.5 mm** | **Root Diameter > 2 mm** | **0.5 mm < Root Diameter > 1 mm** |
| Root Diameter > 2 mm | <0.001 | - | - | <0.001 |
| 0.5 mm < Root Diameter > 1 mm | <0.001 | ns | - |
| 1 mm < Root Diameter > 2 mm | <0.001 | <0.001 | <0.05 |
| | Survival Curve for *Quercus robur* Per Season of Root Birth (with Root Diameter < 2 mm) | | |
| | **Post-Hoc Analysis** | | **Log-Rank Test** |
| | *p* Value | | *p* Value |
| **Season of Root Birth** | **Autumn** | **Spring** | **Summer** |
| Spring | ns | - | - | <0.001 |
| Summer | <0.001 | ns | - |
| Winter | <0.001 | <0.01 | ns |

**Table 4.** *Cont.*

| | Survival Curve for *Betula pendula* Per Root Diameter Class | | | |
| --- | --- | --- | --- | --- |
| | Post-Hoc Analysis | | | Log-Rank Test |
| | *p* Value | | | *p* Value |
| **Root Diameter Class** | **Root Diameter < 0.5 mm** | **Root Diameter > 2 mm** | **0.5 mm < Root Diameter > 1 mm** | |
| **Root Diameter > 2 mm** | ns | - | - | <0.001 |
| **0.5 mm < Root Diameter > 1 mm** | <0.001 | <0.001 | - | |
| **1 mm < Root Diameter > 2 mm** | ns | <0.01 | <0.001 | |
| | Survival Curve for *Betula pendula* Per Season of Root Birth (with Root Diameter < 2 mm) | | | |
| | Post-Hoc Analysis | | | Log-Rank Test |
| | *p* Value | | | *p* Value |
| **Season of Root Birth** | **Autumn** | **Spring** | **Summer** | |
| **Spring** | <0.001 | - | - | <0.001 |
| **Summer** | ns | <0.001 | - | |
| **Winter** | <0.001 | <0.001 | <0.001 | |

## 4. Discussion

### 4.1. The Trends (or Relative Lack Thereof) in the Fine-Root Phenology of Fagus Sylvatica, Quercus Robur, and Betula Pendula

Unlike in the aboveground phenology, none of the species showed a clear and repeating yearly pattern in their fine-root phenology. It is possible that the initial fine-root growth (especially in birch) was the result of soil disturbance and root competition effects, causing a local increase in the nutrient availability after installation of the tubes [3,4,98].

The fine-root phenology in beech and oak (late-successional species) did show more similarities among each other than in comparison with birch (pioneer species). For instance, the mean cumulative root surface in beech and oak showed a clear initial increase around the end of spring and the beginning of summer in 2019 (and also in 2020 for the beech trees in the KS). On the other hand, birch showed a very stable cumulative root surface. The subsequent decrease in the cumulative root surface in the beech (KS) and oak trees (PB) in 2020 might be explained by a higher mortality after the strong initial increase and the stabilization of the root growth dynamics in 2019. On the other hand, for birch in both years, the stable cumulative root surface could indicate that while some fine-roots might be dying, other fine-roots might be compensating for the loss of root surface area by growing in length or width. For beech and oak, other explanations are probably also at hand to explain differences between years. These deal with methodological limitations (e.g., difficulties in determining root death, etc.) or variability in meteorological drivers. The literature suggests a number of environmental drivers affecting the phenology of fine-roots [8,39]. These include mainly extreme soil moisture values, nutrient availability, and low soil temperatures [4,35,128–133]. Premature mortality of fine-roots has also been mentioned as a sign of hydraulic failure, a process which is not an uncommon result of long-lasting heat and drought stress [1,4,11,38,134]. The impact of drought on fine-roots at the study sites is supported by the fact that the decline in the cumulative root surface in beech and oak occurred at the same time when the study sites experienced an exceptionally warm and dry spring (Table 1). This, together with a higher cumulative root surface in 2019, indicates that fine-roots might be more sensitive to drought experienced during spring than drought experienced during summer.

Despite the lack of clear and repeating yearly patterns in the fine-root phenology, it is possible to draw connections between our results and those shown in the literature. For example, it is relatively common for fine-roots to show inconsistent patterns with a wide variability among species and years [4,19,64]. The same literature also repeatedly observed a large variability in the fine-root phenology within one species. For example, Withington, et al. [4] found that the fine-root phenology in beech did not show a peak in some years but showed one peak in March and another peak in summer in other years. The variation in the fine-root phenology within one species is also clear from the difference between the cumulative root surface and the new root count in the beech trees of the KS and PB. We

observed a much higher cumulative root surface in the beech trees of the KS than in those of the PB. Likely, this was due to the difference in the soil types (Zdg vs. W-Zdg) and their respective differences in the soil moisture content. In fact, at the drier KS, trees would likely need more fine-roots than at a wetter PB whenever droughts occur [11].

A second sign of the large variability in the fine-root phenology was our ability to improve the modeling by implementing more-elaborate models that account for the variance, skewness, and kurtosis in the fine-root data. For example, the GAMs with the cumulative root surface as a response variable could be improved by using the GAMLSS distribution. Likewise, the considerable effects of skewed and kurtotic data were noted due to the potential to amend the modeling of the data using GAMLSS models with multiple parameter distributions. For example, the modeling of the cumulative root surface data could be improved by fitting it to a generalized beta type 2 (GB2; four distribution parameters) distribution, while the modeling of the new root count data could improve by fitting it to a zero-adjusted negative binomial (ZANBI; three distribution parameters) distribution [135–139].

Similar to the parabolic pattern in the cumulative root surface in oak that we reported, with a high cumulative root surface from the summer of 2019 to the beginning of spring 2020, López et al. [140] observed the highest fine-root biomass production in *Quercus ilex* L. trees in winter. They explained this phenomenon by stating that fine-roots may undergo less water stress in winter. However, several authors suggest that fine-root growth is limited under a certain soil temperature [36,141–143].

### 4.2. Relationship between the Above- and Belowground Phenology of Deciduous Trees

In 2019, some links between the root phenology and senescence could be observed. For example, a peak in the new root count of the beech trees in the KS and the PB could be observed during or just before the moment with the highest probability on the phenological transition date marking the re-organization phase of senescence, respectively. In the oak trees, the trend in the new root count peaked concurrently with the first peak in the phenological transition data marking the re-organization phase of senescence. However, similar dynamics for beech and oak were not observed in 2020. On the other hand, the fine-roots of birch did show growth during the entire period of senescence in both years. We can observe potential links between the below- and aboveground phenology. Because of the lack of seasonality in the observed fine-root phenology, the coupling between the below- and aboveground phenology was, however, not consistent across years for beech and oak. Studies on the formation, differentiation, and maturation of xylem (i.e., xylogenesis) on the same species also suggested that the relationships between the xylogenesis and the timing of the bud burst or leaf senescence are unclear, or missing [144–147].

In the trend of the cumulative root surface of the beech trees in the KS and PB, one might observe two moments of increase just after the bud burst in 2019 and 2020. The observation that the bud burst in beech occurs later or concurrently with the onset of the wood formation is known from the literature and can be associated with the wood anatomy of beech (i.e., semi-diffuse porous wood). Similar to beech, the bud burst in birch (diffuse porous wood) occurs later or concurrently with the onset of wood formation. The wood formation in oak (ring porous wood), on the other hand, starts before the bud burst [148–150]. However, the observed moments of increase in the cumulative root surface of the beech trees might also sign towards an effect of the re-allocation of resources from different plant organs to the fine-root growth [4,35,133,151–154]. In the literature, internal competition for carbohydrates among plant organs is usually mentioned as a potential explanation for the presence of one- (usually between June and September) or two-yearly peaks in the fine-root phenology of deciduous trees [4,18,19,98,142,155–160]. However, our results mainly suggest that internal processes are less important for fine-root phenology than external drivers. In fact, the lack of parallel observations between the belowground and aboveground phenology may suggest that the phenology of the former is mainly driven by carbohydrate reserves, colonization through mycorrhizae, or meteorological

variables (e.g., soil moisture or temperature) that are different from the drivers of the latter (e.g., air temperature, vapor pressure deficit, etc.) [40,161].

There may be two aspects for which the link between the below- and aboveground phenology may be obvious: the species life strategy (i.e., successional stage) and drought conditions. Relationships between the above- and belowground phenology might depend on the species' life strategy in this study. For example, as a pioneer species, birch has an early bud burst and early onset of its re-organization phase of senescence. Likewise, birch has fine-roots with a shorter median lifespan than those in beech and oak. Perhaps the life strategy of birch may, through a different timing in the internal allocation of carbohydrates, explain why its fine-root phenology is different from those of beech and oak. However, phylogenetic relationships might also be important here as both beech and oak are members of the Fagaceae, but Birch is a member of the Betulaceae. Furthermore, it is known that the root-to-shoot ratio tends to increase under seasonal drought stress, because of a decrease in the fine-root biomass [11,162–167]. In normal conditions, abscisic acid (ABA) is produced in roots, where it will stimulate root growth [50,168]. However, under drought stress, ABA production in roots inhibits root growth and signals the ABA production in the leaves (where it will cause closure of the stomata) [12,50,169,170].

### 4.3. Discussing the Fine-Root Lifespan and Turnover Rate

Although the differences in the overall survival probability were about 10% to 15%, our results showed significant differences in the fine-root lifespan and the turnover rate among species with a different life strategy. Foremost, our results showed that the fine-root lifespan and the turnover rate in beech and oak were greater and smaller, respectively, than those of birch. In fact, our estimations of the fine-root lifespan and the turnover rate in birch deviated notably from the values we expected based on the literature. Our results do correspond to the observation that the fine-roots of late-successional species usually live longer than those of pioneer species [40]. The fine-root lifespan and the turnover rate of beech, a species of which the establishment is considered to be the latest phase in a forest's succession, was even the highest and the lowest, respectively, of all three species.

The results shown in the literature indicate an extremely wide variability in the estimations of the fine-root lifespan and the turnover rate among and within deciduous tree species [13,27,50,53–61,63,66,171]. Nevertheless, our estimations of the fine-root turnover rate in beech, oak, and birch lie remarkably close to the universal fine-root turnover value (0.7 year$^{-1}$) suggested by Hickler et al. [172] for deciduous broadleaf forests. In fact, our estimations corroborate with the fine-root turnover rate (1 year$^{-1}$) usually set in global vegetation models [50,65,173,174].

We found considerable variation in our estimations of the fine-root lifespan and the turnover rate within species, especially whenever the estimations were split by root diameter class or the season of root birth. While we expected roots with a larger diameter to have a greater fine-root lifespan, this was not always the case [3,40]. We did not observe that fine-roots born in spring always lived shorter than those born later [8,32]. In fact, fine-roots born in spring only lived shorter in beech and oak but not in birch, pointing towards a species-specific effect. Nevertheless, we might say that our estimations of the fine-root lifespan were relatively high, given the drought stress [50,175–177]. Unlike for roots with a diameter below 0.5 mm, our estimations of the fine-root lifespan and the root turnover rate in fine-roots with a diameter above 1 mm but below 2 mm also seemed to be rather stable among species.

### 4.4. Study Limitations

Measuring fine-roots using the minirhizotron is labor intensive and, as with any other technique to assess fine-roots, prone to limitations (see Section 1.2) [3]. Even the material of the minirhizotron tubes might have affected the fine-root phenology, lifespan, and turnover rate estimations, as Withington et al. [178] showed that minirhizotron tubes made of butyrate, in comparison to those made of glass or acrylic, negatively affected the

survivorship of fine-roots in hardwood tree species. Due to the monthly sampling sessions and the difficulty in assessing the status (alive or dead) of the fine-roots, we expected inaccuracy in the measurements of the fine-root lifespan and the turnover rate. In fact, more-accurate estimates of the fine-root lifespan and the turnover rate likely require longer time series that are more frequently observed [19]. In addition, it is noteworthy that the $R^2$ (ranging from 0.11 to 0.28) was relatively low in all GAMs. It is likely that this was due to the relatively long periods between sampling and the inherent difficulties related to minirhizotron measurements (e.g., the effect of soil movement, tube instability, installation, or simultaneous growth and dying of the root, only a small area being observed, etc.). The GAMs should therefore be interpreted with caution. Small differences in the soil moisture between the KS and PB might also have affected the fine-root phenology but could not be fully accounted for.

## 5. Conclusions

We summarize here the answers to our three research questions:

(I) How do the below- and aboveground phenology of deciduous trees relate to each other?

We observed no clear and repeating yearly patterns in the belowground phenology of beech, oak, or birch. Consequently, no consistent coupling between the below- and aboveground phenology of these species could be observed. In 2019, we observed peaks in the new root count concurrently with the phenological transition date marking the (re-organization) phase of senescence in beech and oak. Our data, which were asymmetrically distributed with strong effects from outliers, suggest that the life strategy of a species influences both the below- and aboveground phenology and the fine-root lifespan and turnover rate values.

(II) Are there differences in the fine-root lifespan, turnover rate, and phenology among beech, oak, and birch?

The fine-root phenology in beech and oak (both late-successional species and members of the Fagaceae) showed more similarities among each other than in comparison with birch (pioneer species and member of the Betulaceae). Likewise, our results also showed significant differences in the fine-root lifespan and the turnover rate of beech, oak, and birch. More specifically, beech and oak had the longest fine-root lifespan and the lowest turnover rate, respectively.

(III) Are there differences in the fine-root lifespan and the turnover rate among roots with different diameters or among those born in different seasons?

Fine-roots with a larger diameter did not always have a greater fine-root lifespan than fine-roots with a smaller diameter. Fine-roots born in spring also did not always live shorter than fine-roots that were born later. Significant differences, and a wide variance in the estimations, were especially found in the fine-root lifespan and the turnover rate whenever the fine-root data were divided by the root diameter class or the season of root birth.

The yearly pattern in the phenology of the fine-roots was unclear with a wide variance in the cumulative root surface and the new root count. It is likely that the lack of clear and repeating phenological trends, and the wide variance in the fine-root lifespan and the root turnover rate estimations, might have been due to the extreme heat and drought stress observed during the period of 2019–2020.

**Author Contributions:** Conceptualization, methodology, and funding acquisition, M.C.; investigation, H.X.N., C.F., T.G., P.W. and B.M.; formal analysis, B.M.; validation, P.S. and I.O.; visualization, B.M.; writing—original draft, B.M.; writing—review & editing, A.P., H.X.N., P.S., P.W., I.O., M.C. and B.M. All authors have read and agreed to the published version of the manuscript.

**Funding:** B.M. and M.C. have been funded by a DOCPRO4 fellowship and the ERC Starting Grant LEAF-FALL (714916) by the University of Antwerp and the European Research Council, respectively. I.O. and P.S. (in part) were supported by Estonian Research Council grant PRG916.

**Data Availability Statement:** Data will be made available upon acceptance when required.

**Acknowledgments:** Thanks are due to Sebastien Leys (PLECO), the Flemish Agency for Forest and Nature (ANB), the Belgian Armed Forces, and the Municipality of Brasschaat.

**Conflicts of Interest:** The authors declare no conflict of interest.

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
