# Peer review of "On the Below- and Aboveground Phenology in Deciduous Trees: Observing the Fine-Root Lifespan, Turnover Rate, and Phenology of Fagus sylvatica L., Quercus robur L., and Betula pendula Roth for Two Growing Seasons"

_forests, doi:10.3390/f12121680_

Round 1
Reviewer 1 Report
This is a good study reporting valuable field observations. I have no major concerns. Please see some minor comments below.
- Abstract: Never mentioned the relationships between below- and above-ground phenology. I think the results suggested that seems there is “no” relationship between these. But given the title of the manuscript, I think it should be mentioned in the abstract.
- The three questions (L139-143): For me, it was unclear, and couldn’t find the answers to these questions. Some of them were mentioned in the discussion, but these were vague. Please revise associated descriptions in the discussion, especially focusing on new findings from the study.
- Some of the graphics have low quality. Please make sure everything can be seen clearly.
Reviewer 2 Report
Review of the submitted manuscript entitled On the below- and aboveground phenology in deciduous trees: observing the fine-root lifespan, turnover rate and phenology of Fagus sylvatica, Quercus robur and Betula pendula for two growing seasons
The research described in the manuscript concerns a very interesting and poorly known phenology of tree roots. This research certainly contributes to the understanding of tree physiology and has potentially high usefulness for forest management. However, studies on three species are insufficient to infer phenological patterns between early and late accession species.
Generally, the manuscript is very well written. The introduction is informative. The aim of the research and the hypotheses are scientifically justified. Regarding the methodology, I only have reservations about the selection of study sites, which may impact the results and their interpretation. I have no objections to the statistical methods. I like the presentation of the results obtained.
However, it might be worth trying to answer here to what extent the phenology of the roots results from the habitat conditions and whether there are any differences between individuals. I am afraid this may be less clear-cut than the authors describe.
Specific comments:
L.148-153: The research was carried out on three phenologically, physiologically and ecologically different species in two locations. The slight distance does not seem to significantly impact the differences in meteorological conditions between the two locations. However, I am concerned that differences in soil and moisture conditions may influence root phenology. In my research on wood formation dynamics, the differences in cambium activation reached several weeks in three equally-aged pine stands, several hundred meters apart (in preparation). However, they were of the same type. There could have been undetectable differences in humidity, for example. In the studies described in the manuscript, beech and oak grow in different sites! So there are even more factors that can influence the results.
L.583-599: In my opinion, studies based on only three species are insufficient to draw unambiguous conclusions regarding the differences in root phenology between early and late accession trees.
Each of these species differs significantly in terms of anatomical structure, phenology and physiology.
Birch is a typical early-succession species with expansive wood, in which the activity of cambium in the trunk begins together with the development of leaves. The same is valid for beech, which has diffuse-porous or semi-ring-porous wood. The situation is quite different in ring-porous oak, which flushes leaves up to a few weeks later after activating the cambium in the stem. In all study species, activation of the cambium in the roots occurs later than in the stem.
The differences between diffuse-porous and ring-porous trees in leaf and stem phenology and carbohydrate dynamics probably strongly influence the root phenology. I am concerned that generalizing the differences between early- and late-season species requires more species research, see:
Michelot, A., Simard, S., Rathgeber, C., Dufrêne, E., & Damesin, C. (2012). Comparing the intra-annual wood formation of three European species (Fagus sylvatica, Quercus petraea and Pinus sylvestris) as related to leaf phenology and non-structural carbohydrate dynamics. Tree Physiology, 32, 1033–1045. https://doi.org/10.1093/treephys/tps052
I am still wondering what are the relationships between aboveground and underground phenology, since recent studies in leaf and cambium phenology have shown no clear relationship, see:
Sass-Klaassen, U., Sabajo, C. R., & den Ouden, J. (2011). Vessel formation in relation to leaf phenology in pedunculate oak and European ash. Dendrochronologia, 29, 171–175. https://doi.org/10.1016/j.dendro.2011.01.002
Puchałka, R., Koprowski, M., Gričar, J., & Przybylak, R. (2017). Does tree-ring formation follow leaf phenology in Pedunculate oak (Quercus robur L.)? European Journal of Forest Research, 136(2), 259–268. https://doi.org/10.1007/s10342-017-1026-7
Would you mind discussing this?
In my research, I found the lack of patterns repeating every year between the phenology of leaves and cambium phenology in oak and beech (in preparation). So, the lack of repeatability of the phenological patterns in the case of the roots does not surprise me here. I also do not find a simple explanation here. However, I believe that this is an interesting thread for further research.
Kind regards
Round 2
Reviewer 2 Report
The authors provide suggested corrections to the manuscript. Currently, the article requires major technical corrections, as listed below.
Statistical formulas are hardly readable.
Latin names for species should always be spelt with italics.
L.32-33: The task of the keywords is to provide additional information about the content of the article not included in the title. Their selection is crucial because it affects the positioning of publications in web browsers and their search.
If there are repeated words from the title, they lose their function. Here, I suggest you replace Latin generic names with English names.
L.361-365: When citing literature, the first author, et al., and the reference number in square brackets. This also applies to the rest of the manuscript.
L.750-753: Studies on the formation, differentiation and maturation of xylem (i.e. xylogenesis) on the same species also suggested that a clear association between the xylogenesis and the timing of the bud burst or leaf senescence is unclear, or missing [145-148]. -> Studies on the same species also suggested that the relationships between the xylogenesis and the timing of the bud burst or leaf senescence are unclear or missing [145-148].
L.880-886: Please also notice that beech and oak belong to the same plant family. Hence, the similarity may also be related to phylogenetic relations. Hence, the similarity may also be related to phylogenetic relationships.
I am not a native speaker, but I feel that language correction is strongly needed for this manuscript.
